# Investigation of Archaeological European White Elm (*Ulmus laevis*) for Identifying and Characterizing the Kind of Biological Degradation

**Amir Ghavidel [1],\*, Jana Gelbrich [2], Aldi Kuqo [3], Viorica Vasilache [4] and Ion Sandu [4]**

[1]  Doctoral School of Geosciences, Alexandru Ioan Cuza University of Iasi, 700506 Iasi, Romania
[2]  Leibniz-IWT—Institute for Materials Testing, 28359 Bremen, Germany; gelbrich@mpa-bremen.de
[3]  Wood Biology and Wood Products Burckhardt Institute, Georg-August-University Göttingen, 37073 Göttingen, Germany; aldi.kuqo@forst.uni-goettingen.de
[4]  Institute of Interdisciplinary Research-Field Science, Alexandru Ioan Cuza University of Iasi, 700107 Iasi, Romania; viorica.vasilache@uaic.ro (V.V.); ion.sandu@uaic.ro (I.S.)
\*  Correspondence: amir.ghavidelesfahaln@student.uaic.ro; Tel.: +46-734-395-973

**Abstract:** The current work aims at the study of the biological degradation of archaeological European white elm via microscopy and chemical analysis in order to identify the kind of biological degradation and characterize the state of preservation of this type of wood. Profound knowledge of the chemical constituents and biological degradation in fresh-cut and archaeological elm wood will simplify the process of restoration and conservation of the investigated artifacts. Therefore, fresh-cut and archaeological elm were compared in terms of extractive, chlorite holocellulose, α-cellulose, lignin, and ash contents. In the fresh-cut elm wood, the contents of Kürschner–Hoffer cellulose, chlorite holocellulose, α-cellulose, and hemicellulose were significantly higher than that of the archaeological elm, confirmed by the degradation of native wood hemicelluloses by erosion bacteria during soil contact. Naturally, the mass percentage of lignin increases as the amount of chlorite holocellulose in the wood decreases. These wet chemistry results were also confirmed by FTIR analysis, where bands mainly attributed to hemicellulose and cellulose decreased significantly and bands belonging to lignin display higher intensity for the archaeological specimens. Ash and cyclohexane–ethanol extract contents of archaeological elm wood were significantly higher due to the movement of mineral components arising out of the soil into the wood specimens. Based on the microscopic investigation and given the fact that wood decay fungi need oxygen to degrade wood and the investigated archaeological elm specimens were buried to a 10 m depth in the soil, we might conclude that the wood degradation was caused by erosion bacteria.

**Keywords:** archaeological wood; elm wood; FTIR-ATR; biological degradation; erosion bacteria

## 1. Introduction

Wood is a naturally robust material long known for its flexible and attractive structural properties and engineering. In contrast to the misapprehension that wood is not durable, wood can last for centuries in a favorable climate [1,2]. Wood items, aged from the 14th century, were found in perfect condition in the Egyptian pharaoh Tutankhamun's tomb when recovered in the 20th century. Several Japanese, wood-built temples date back 13 centuries. In another case, a covered bridge built in 1440 is still in service in Lucerne, Switzerland, and some 950 covered bridges constructed during the 19th century remain in service in the United States.

However, wood is susceptible to environmental biodegradation as with other biological materials. Wood biodegradation is an essential part of the carbon cycle. In general, in the decomposition of

wood, a great number of different species are involved, such as insects, calms, nematodes, fungi, and bacteria. The growth and development of each of the aforementioned species has its own role in this complex system [3]. Wood is rapidly decomposed by microorganisms in most terrestrial environments, releasing carbon dioxide, water, and minerals. There are many aggressive wood-destroying fungi, such as white and brown rot fungi. Such basidiomycetes specialize in colonizing wood and extensively degrading lignocellulosic materials under certain conditions of the environment and the substrates. Still, most terrestrial ecosystems usually have their optimum conditions [4]. Another group of fungi that includes ascomycetes and deuteromycetes species are capable of attacking wood exposed in the soil in very high moisture contents, as well as in fresh and saltwater. These soft rot fungi are considered to have a lower oxygen demand than basidiomycetes [3]. More or less, humid conditions and air contact are essential for all wood-destroying fungi since the fungal enzyme system uses oxygen as an electron acceptor in the process of decomposing the complex wood structure.

Many fungi can degrade modified cellulose products but only a limited number of so-called cellulolytic fungi can decompose the native, highly crystalline cellulose. These have a full enzyme system for extracellular degradation of crystalline cellulose with endo- and exocellulose and usually belong to ascomycetes, deuteromycetes or basidiomycetes [5].

Besides wood rot fungi, bacteria can degrade wood also in such a way they can experience significant losses in strength. Three types of wood degrading bacteria are described in literature based on their specific micro-morphological features: Tunneling bacteria [6], erosion bacteria [7], and cavitation bacteria [8]. Research showed that tunneling bacteria (TB) can be found in environments similar to soft rot conditions, while erosion bacteria (EB) are able to degrade wood in environments characterized by a very restricted oxygen supply [9]. They appear to be most tolerant near to anoxic conditions [10–13], on the other hand, the main wood degraders under waterlogged conditions [13–15]. Experiments even suggested that EB can be involved in wood degradation even without free oxygen present, however the process is more intense if oxygen is available [13].

The purpose of this study was the investigation of biological degradation of archaeological European white elm (*Ulmus laevis*) originating about 2000 years ago and found in Vetiș (Romania) buried at approximately 10 m depth. The characterization of biological degradation was done by light microscopy as well as chemical analysis and FTIR spectroscopy. In this study, scanning electron microscope (SEM) imaging was used to show the structure of archaeological and fresh-cut elm wood. Such samples have resisted severe conditions in the wet ground for around 2000 years; hence, their conservation status is little affected.

## 2. Materials and Methods

### 2.1. Raw Materials and Preparation of Samples

The fresh-cut specimens of elm wood (*Ulmus laevis*) were provided by a local sawmill. These consisted of boards with dimensions 45 × 90 × 4000 mm that were obtained from Suceava region, Romania's most important grown forest areas. The specimens were taken out of heartwood. Since knots and other abnormal parts of wood contain the highest amount of lignin and the lowest amount of cellulose compared to other zones, for our tests, defect-free specimens (no knots, cracks, or wood reaction, and annual growth rings slope <5°) were selected. The archaeological elm wood specimen was extracted from sand mines. Specimens were located in the alluvial layer of more than 10 m thickness from the former riverbed. Some of them are stationed in the city of Vetiș county of Satu Mare-Romania. This wood specimen was buried at a depth of 8–10 m and its age varies from 1800 to 2000 years. The type of usage is unclear. The anatomical structure of this specimen was completely altered for a long time due to the environmental factors such as humidity and temperature as well as biologically degrading agents such as fungi and bacteria of the wood. The anatomical structure can also be considerably affected by the sample's burial in the soil.

Specimens of wood were subjected to chipping and drying at 105 ± 3 °C in an oven. Then, dried specimens were ground and sieved to a powder. For all chemical analyses, the fraction 250–425 μm was used.

## 2.2. Analysis and Evaluation Methods

### 2.2.1. Microscopic Imaging

Transverse sections of the archaeological and fresh-cut elm wood specimens were prepared manually by using a cutting tool; test pieces were kept in wet conditions during sectioning. Sections of wood were stained for 10 min with 0.1% safranin, then detained by washing with 50% ethanol for approximately 15 min. In the last step, the specimens were washed using distilled water, placed on a microscope slide, and covered with a cover slip. Specimens were imaged with a DS-Fi1c high-definition cooled color camera and Nikon ECLIPSE E600 microscope equipped with the NIS-Elements software (Nikon, Tokyo, Japan).

### 2.2.2. Moisture Content

The moisture content of wood specimens was assessed by using an Ohaus MB23 moisture analyzer. In our case, the moisture analyzer measures moisture thermogravimetrically. Approximately 1 g of each sample were placed into the device. Thermogravimetric moisture analysis defines moisture as the loss of mass observed when the sample is heated and is based, in theory, on the vaporization of water during the drying process.

### 2.2.3. Cyclohexane-Ethanol Extract

A cyclohexane: ethanol 50:50 (*v/v*) solution (100 mL) was used for the extraction process. Specimens weighing 2–3 g were placed into sealed extraction thimbles and extracted with 100 mL of solution using Soxhlet extractor for 24 h. Following the extraction, the extract was put in a 100 mL volumetric flask and filled with a pure solvent (ethanol, 99.99%). Then, 20 mL was pipetted to a plastic tray from these precisely filled solutions and allowed to dry at room temperature. The amount of the extract was then weighed. For the measurement of the cyclohexane-ethanol extract, the following formula was used:

$$X(\%) = 100 \times 5 \frac{(T2 - T1)}{G \frac{(100 - N)}{100}} \tag{1}$$

where $T1$ is the weight of the empty plastic tray; $T2$ is the full weight of the plastic tray with dry extractives, $G$ is the sample weight; and $N$ is the moisture content of samples. Analyses were repeated three times.

### 2.2.4. Chlorite Holocellulose Content

After the extraction (process described in Section 2.2.3.), wood specimens were put into an Erlenmeyer flask. Thereafter, a solution composed of distilled water (80 mL), acetic acid 96% and 1 g of sodium chloride was prepared. The solution was heated in a water bath for 1 h in 70 °C. Portions of 0.5 mL of acetic acid and 1 g NaCl were added after each consecutive hour. This step was repeated 6 times. Then, samples were left overnight in the water bath. After cooling, they were filtered on a G1 glass funnel filter (at the end of the 24 h reaction) and the chlorite holocellulose was washed with 10 mL acetone and water until the yellowish color was removed [16]. The chlorite holocellulose was subjected to drying at 105 °C and weighting.

### 2.2.5. α—Cellulose Content

The previously prepared chlorite holocellulose was placed into a 250 mL glass beaker, and 5 mL portions of a 17.5% NaOH solution were added at 5 min intervals until the solution was entirely

filled. The response time was 1 h. Filtered on a G2 porosity glass funnel filter, the $\alpha$-cellulose was washed with acetic acid, 5% NaOH, and water. The washing process was repeated twice. The prepared $\alpha$-cellulose was subjected to drying at 105 °C and then weighed [16].

### 2.2.6. Kürschner–Hoffer Cellulose Content

The Kürschner and Hoffer method [17] was used in this study to obtain the cellulose content. The detailed procedure is described as follows: 1 g of the sample was refluxed with 100 mL of a solution, prepared of nitric acid: ethanol 20:80 *v/v* for 1 h. Once the reaction solution was completed, digestion followed and was repeated for 1 h with the same mixture. The prepared cellulose from the Kürschner–Hoffer was filtered using a G2 glass funnel filter and washed with 10 mL ethanol and 500 mL hot distilled water.

For the aforementioned tests, the following formula was used:

$$X(\%) = 100 \times 5 \frac{(T2 - T1)}{G \frac{(100-N)}{100}} \tag{2}$$

where $T1$ is the weight of the empty filter funnel; $T2$ is the weight of the full filter funnel; $G$ is the weight of samples; and $N$ is the moisture content of each sample. Analyses were carried out three times.

### 2.2.7. Lignin Content

Total lignin was estimated by the remaining fraction of wood samples after the determination of chlorite holocellulose, ash content, and cyclohexane-ethanol extract by using the following expression:

$$L = 100 - A_{db} - HC - E \tag{3}$$

where $A_{db}$ is the Ash content, $HC$ is the content of chlorite holocellulose, and $E$ is the amount of extractives.

### 2.2.8. Ash Content

The determination of ash content was performed according to the EN 15.403 standard [18]. Similar particle sizes used in chemical analysis have also been used in the present procedure. The wood samples were dried at 105 ± 3 °C by using an oven and portions of approximately 2.5 g were put to pre-dried ceramic dishes. The dishes were transferred in the cold furnace, over a period of 50 min, and the furnace temperature was increased constantly to (250 ± 10) °C. This level of temperature was kept for 60 min so volatiles could evaporate from the sample prior to ignition. Over a period of 60 min, the furnace temperature continued to rise constantly to (550 ± 10) °C and this level of temperature was kept for 120 min. The dishes and their contents were removed from the furnace, left to cool for 5 to 10 min on a thick metal plate, before being put to a non-desiccant desiccator and allowed to reach the room temperature. The dishes were weighed after 30 min. The ash content was calculated using the following equation:

$$A_{db} = \frac{(m_3 - m_1)}{(m_2 - m_1)} \times 100 \tag{4}$$

where $m_1$ is the mass of the empty dish; $m_2$ is the mass of the dish plus the general sample; and $m_3$ is the mass of the dish with the ash. Analyses were carried out three times.

### 2.2.9. FTIR Analysis

Prismatic specimens were cut from each wood with an edge length of 5 mm and a thickness of 1 mm. On the Bruker Invenio R spectrometer fitted with a diamond ATR unit (Bruker Optik GmbH, Ettlingen, Germany), the ATR FTIR spectra were recorded in the range of 4000–400 cm$^{-1}$ using 64 scans at a resolution of 4 cm$^{-1}$. To ensure reproducible and constant force, the specimens were pressed

onto the ATR-crystal with an integrated applicator. A background spectrum with an empty specimen compartment was recorded prior to measurements and automatically subtracted from the spectra in the measurements below. For each wood, two specimens were analyzed, and their spectra were averaged. Based on the software OPUS version 8.2 (Bruker Optik GmbH, Ettlingen, Germany), the spectra were baseline corrected, and the vector was normalized.

### 2.2.10. Characterization by Scanning Electron Microscope (SEM)

For archaeological and fresh-cut elm wood specimens $10 \times 10 \times 10$ mm$^3$ sized specimens were used. The surface was cut with a razor blade before imaging. No metal coating prior to specimen imaging was applied for wood specimens. A TESCAN MIRA3 equipment was used for SEM imaging, using a 70 Pa vacuum and a 15 kV accelerating voltage.

### 2.2.11. Statistical Analysis of the Results

Distribution normality of the data was verified and statistical significance tests (ANOVA, Fischer LSD-test, $p < 0.05$) were conducted on the investigated material properties with the software Statistica 13.0 (Statsoft).

## 3. Results and Discussion

### 3.1. Evaluation of Bacterial Attack by Light Microscopy

Figure 1 shows SEM images of fresh-cut and archaeological elm wood species, observation of the collapsed cell structures indicated that archaeological wood has experienced an obvious compression.

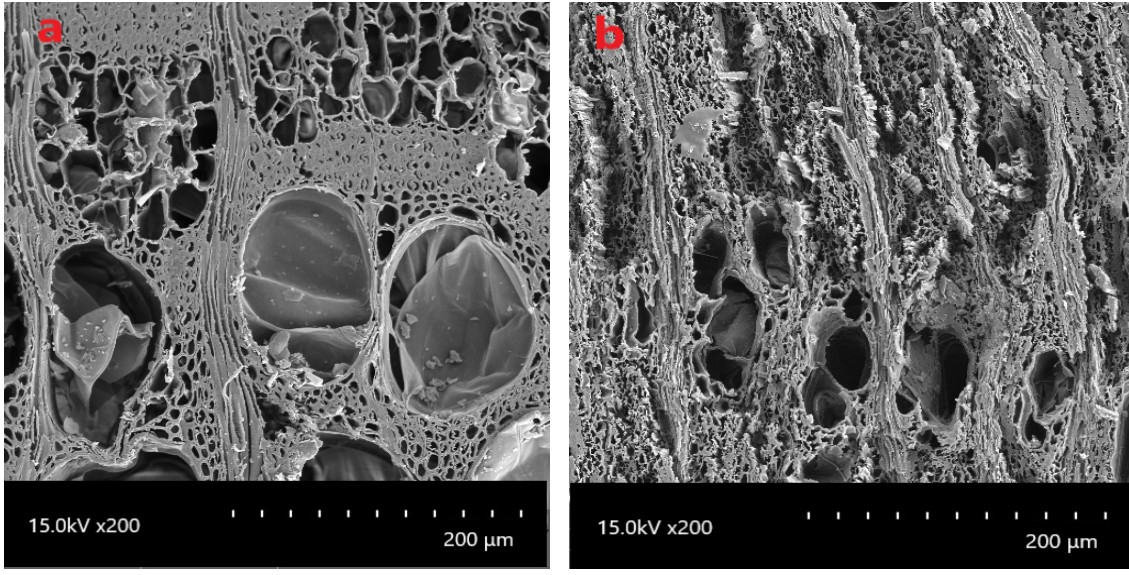

**Figure 1.** SEM microphotography of anatomical elements of wood: (**a**) Fresh-cut; (**b**) archaeological elm wood.

Light microscopy of wood sections revealed the characteristics of bacteria in the archaeological elm wood specimen. No signs of hyphae or characteristic soft rot patterns could be found. Advanced stages of decay were present. Because archaeological elm specimens were buried 10 m deep in the soil where likely a low amount of oxygen is present, there is a possibility of bacterial degradation of historic elm specimens. Figure 2 demonstrates the possibility that bacteria can colonize and degrade the lignified cell walls of timber buried in low oxygen soil as described in the literature [19–24]. Light microscopy of hand-cut sections of historic elm wood reveals a distinctive pattern of bacterial erosion. The current results on cell wall degradation are based on observations of EB-attacked wood

obtained from different environments and conditions, including archaeological sites and laboratory experiments [25,26]. The pattern of degradation was similar in all cases. In the longitudinal direction, a lot of eroded, thin channels (troughs) can be seen in an oriented way, aligned along the cellulose microfibrils. In transverse sections, the erosion starts from the lumen and leads to erosion troughs in the wood cell wall around the bacteria [3,4,14,27]. It is already known that hemicelluloses are usually first degraded, followed by cellulose, hardwood lignin, and then softwood lignin [28]. In this study, the results of chemical analysis on a specimen of archaeological elm wood confirm this result. Prolonged exposure of wood to low-oxygen conditions can cause significant losses in polysaccharides, which can also completely deplete the wood holocellulose [27].

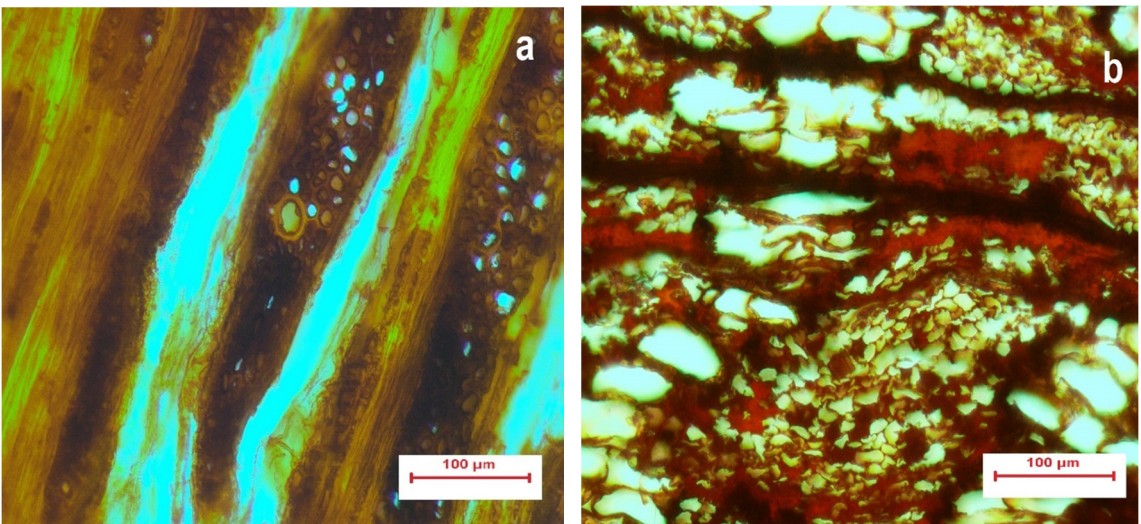

**Figure 2.** Microscopic image of archaeological elm specimens degraded by erosion bacteria: (**a**) Tangential section; (**b**) transversal section.

The secondary wall of axial tracheids and ray parenchyma is degraded for the first time, whereas the middle lamella is mostly intact even in the cells with the higher degree degradation [29]. Singh et al. [29], while investigating the archaeological wood of Pinus sylvestris, showed that various wood cell types or cell walls layers have different resistance to bacterial attack.

In some cells, the pit borders in axial tracheids were not entirely but more resistant to degradation than the rest of the secondary walls [30]. Pit borders are comparable in chemical composition to the rest of the secondary cell wall, but their accessibility to erosion bacteria (EB) may have been limited. A warty layer, very resistant to chemical dissolution due to the high lignin content, completely covered the interior of pit's borders. A dense layer, likely to be rich in extractives and toxic to microorganisms, coated the warty layer in the archaeological wood [29]. The widely accepted view that primary structures are the initial pit boundaries, would indicate that both have almost the same lignin material. That would clarify EB's resistance to degradation of the initial pit boundaries [29,31].

It is known that the ray tracheid walls in archaeological wood are completely invulnerable. The high lignin content of ray tracheid walls, and the presence of an extractive layer, which can prevent bacterial entry into the ray tracheids, are the main contributors of this increase in resistance [29].

EB has minimal capacity to degrade lignin-rich wood cell wall areas but is significantly larger for soft rot fungi [32]. The P. radiata S3 layer with a 52% lignin content tends to be resistant to soft rot fungi but only moderately resistant to EB [33].

### 3.2. Chemical Analysis

The results of the chemical analysis in fresh-cut and archaeological specimens of elm wood are summarized in Table 1. Because there are many different ways of determining cellulose from wood,

and because the results of these methods are not entirely precise, two different methods were used to test the cellulose material. The Kürschner–Hoffer (1929) process is based on the process of refluxing of wood with nitric acid for 1 h ethanol mixture 20:80 *v/v*. The experiment is quick and easy to carry out compared to other methods; however, cellulose prepared by the Kürschner–Hoffer method may include hemicellulose residues, and often this method damages some parts of cellulose [16]. The other approach is the chlorite holocellulose test combined along with the determination of α-cellulose. In addition, longer reaction times may result in an excess of chloritization and loss of some of the polysaccharides [16]. In the past few years, researchers have used different analysis methods for the determination of structural carbohydrate contents of wood and fossil wood compared to the Kürschner–Hoffer method.

**Table 1.** Results of the chemical analysis in fresh-cut and archaeological elm. Superscript letters at $p < 0.05$ level display significant differences between archaeological and fresh-cut specimens.

| Samples | Kürschner-Hoffer Cellulose (wt%) | Chlorite Holo-Cellulose (wt%) | α-Cellulose (wt%) | Hemi-Cellulose (wt%) | Lignin (wt%) | Cyclohexane-Ethanol Extract (wt%) | Ash (wt%) |
|---|---|---|---|---|---|---|---|
| Fresh-cut | 60.84(±0.98) [a] | 82.17(±1.30) [a] | 56.90(±0.60) [a] | 25.94(±0.24) [a] | 14.60(±0.36) [a] | 1.38(±0.01) [a] | 0.66(±0.01) [a] |
| Archaeological | 36.13(±0.77) [b] | 47.91(±1.08) [b] | 32.85(±0.34) [b] | 16.60(±0.13) [b] | 44.22(±0.53) [b] | 2.16(±0.02) [b] | 5.37(±0.04) [b] |

As specified by Rowell (2012), the content of chlorite holocellulose in Ulmus species is 73% and can vary with age and location of the investigated tissue. In this study, the amount for fresh-cut elm wood specimens was 82.17% and for the archaeological one was 47.91%. The differences are significant. Pettersen [34] claimed that α-cellulose content of Ulmus species varies between 47–51%. There is an analogy of Kürschner–Hoffer cellulose and α-cellulose results as significant differences were observed between fresh-cut and archaeological elm wood. These findings indicate that there may have been changes to other wood components during wood aging. According to Pettersen [34], the cellulose content of *Ulmus* species varies at 56–62%. Krutul et al. [35], on the other hand, reported that cellulose contents vary 32.1–35.2% (Kürschner–Hoffer method) for archaeological oak wood.

The content of hemicellulose in the archaeological elm wood specimen was significantly lower compared to the fresh-cut one. According to the literature, the amount of hemicellulose is 25% for hardwoods [16]. This finding can be justified by hemicellulose degradation as the second category of degraded materials during wood ageing in soil contact [16]. Because of the relative decrease in chlorite holocellulose, the content of lignin in archaeological woods was higher than in the reference species [36]. Pettersen [34] reported that the amount of hemicellulose for hardwoods was 23 ± 3%. According to Baar et al. [37], the measured lignin content in the subfossil oak wood specimens was reported to be at 28.9–31.5%, while Krutul et al. [35] reported fossil wood lignin content at 39.3–40.8%. The cyclohexane-ethanol extract of the archaeological wood specimens was significantly lower compared to the respective extract of fresh-cut specimens. It can be assumed that this is caused as a result of leaching of extractives during ground contact in the initial phase of wood aging. [34]. The ash content in the archaeological wood specimen was 5.37%, while in the fresh-cut specimen it was only 0.66%. Ash content in the historical elm wood is significantly higher thanks to the movement of inorganic elements arising from the soil [34,38].

*3.3. FTIR Analyses*

FTIR spectroscopy has been applied to identify changes in the chemical structure of cellulose, hemicelluloses, and lignin due to ageing of elm wood.

Figure 3 shows the ATR FTIR spectra of fresh-cut and archaeological elm wood specimens. It is worth noting that the value of the standard deviation varies throughout the spectrum for the archaeological specimen. Relatively high absorption is seen around bands that are susceptible to shift caused by aging and biological degradation. In archaeological wood, a slight decrease in intensity is observed around 3343 cm$^{-1}$ assigned to OH groups. Changes in OH vibrational bands can be caused

by the absorption of ambient moisture from wood specimens during measurement. The shape of the reference spectrum varies from the spectra of the archaeological specimens in the region between 3000 and 2800 cm$^{-1}$, which indicates changes in the aliphatic part of the molecules that cause band shifts. The peak at 1734 cm$^{-1}$ belonging to the C = O group in hemicelluloses in archaeological specimen does exhibit a significant decrease in the intensity to the reference spectrum. Cellulose and hemicellulose are the first elements that are degraded during aging and with their degradation, the percentage of lignin increases [39]. Also, between 1645 and 1237 cm$^{-1}$ bands of adsorbed OH, β-glucosidic bonds or conjugated C = O groups display high intensity for the archaeological specimen. The increase in bands between 1645 and 1237 cm$^{-1}$ belonging to the lignin might be caused by the leaching of hemicellulose and cellulose in the naturally aged wood [19]. At 1158 cm$^{-1}$ the absorbance band assigned to COC bridge, O stretching vibration in cellulose, lignin, and xylan is seen. These bands, together with 1121, 1051, and 1028 cm$^{-1}$, mainly attributable to CO vibrations in hemicelluloses, decrease significantly for the archaeological elm specimen. Those changes can be attributed to cellulose and hemicellulose degradation through aging processes and specially in our work due to the bacterial degradation [39].

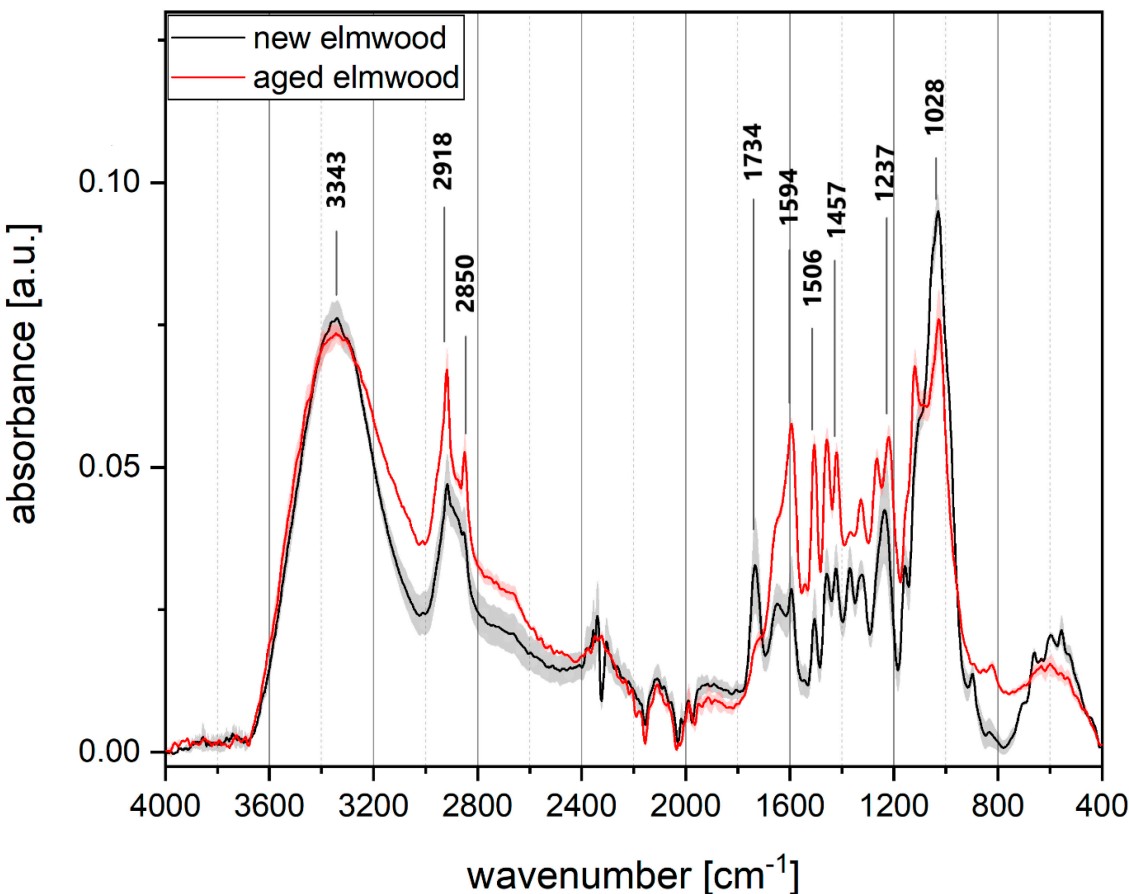

**Figure 3.** FTIR spectra of fresh-cut and archaeological elm wood specimens. The semi-transparent area around the data plot indicates the standard deviation of the two averaged spectra.

The results by FTIR in this study of fresh-cut and archaeological buried stored elm wood are comparable to FTIR results to evaluate bacterial degradation in softwood [40].

## 4. Conclusions

Current research study on fresh-cut and historic elm (*Ulmus* spp.) shows that the contents of Kürschner–Hoffer cellulose, chlorite holocellulose assays/α-cellulose, and hemicellulose in the fresh-cut wood were significantly higher than that of the archaeological one, which was explained

by the degradation of native wood hemicelluloses during soil contact by erosion bacteria and most likely in the low oxygen amount. Compared to the fresh-cut elm wood, the amount of lignin in the archaeological elm wood grew comparatively. These data can be used for archaeometric purposes. Due to the incorporation of inorganic elements from the soil into the wood, the ash and cyclohexane–ethanol extract contents of the archaeological elm wood were significantly higher. The following bands, 1121, 1051, 1028 cm$^{-1}$, mainly attributable to CO vibrations in hemicelluloses, decrease significantly for the archaeological elm specimen and between 1645 and 1237 cm$^{-1}$ adsorbed OH bands, β-glucosidic bonds or conjugated C = O groups display high intensity for the archaeological specimen, according to the ATR FTIR spectrum. Such results indicate that cellulose and hemicellulose are the main elements to be degraded during storage completely buried and that the percentage amount of lignin increases with their degradation. For several metabolic reactions, wood-degradation fungi are known to use molecular oxygen as an electron acceptor. On the other hand, a variety of bacteria involved in the primary degradation and humification of organic matter can colonize wood that is exposed to soil and low oxygen. Considering that archaeological elm specimens were stored in the soil to a depth of 10 m and the possibility of being in low oxygen conditions, the tendency of destruction by erosion bacteria is higher than wood-degradation fungi. Bacterial degradation of archaeological elm wood resulted in a decrease in the chlorite holocellulose content and, thus, an increase in the relative percentage of lignin by residual enrichment.

**Author Contributions:** Conceptualization, A.G. and J.G.; methodology: A.G., J.G., A.K. and V.V.; validation: A.G., J.G. and I.S.; investigation: A.G., J.G.; resources: A.G., J.G. and I.S.; writing—original draft preparation: A.G. and A.K.; writing—review and editing: all authors. All authors have read and agreed to the published version of the manuscript.

**Funding:** This research received no external funding.

**Conflicts of Interest:** The authors declare no conflict of interest.

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
