# Peer review of "Investigation of Archaeological European White Elm (Ulmus laevis) for Identifying and Characterizing the Kind of Biological Degradation"

_heritage, doi:10.3390/heritage3040060_

Round 1
Reviewer 1 Report
Very interesting paper. I've listed below my suggested revisions. I also suggest to carefully re-read the manuscript to correct typos and minor English grammar errors.
Abstract, line 23: I suggest the word assays is removed.
Abstract, Line 27: I believe there is a missing word in this sentence. Please verify, perhaps the following correction is needed: “also confirmed by FTIR”
Introduction, line 40: Pleas correct the typo, it should be 14th century.
Introduction, line 47: Suggest replacing the word included for involved.
Introduction, line 48: It is not clear what is meant by the following sentence: “Growing of each one has its own role in this complex system.”
Introduction, lines 76-77: Please paraphrase the last sentence to clarify what is meant by conservation status. Do you mean that a non-destructive assessment was not needed due to the specimens being already decayed? Also, I suggest, the word stored is removed from this sentence since the samples were buried at a depth of 10m but not necessarily stored in soil.
Materials, line 82 – 83: Please elaborate why it was necessary to compare the cellular structure of freshly cut elm wood that had no defects. How would the presence of reaction wood, or a steep annual growth ring change the anatomical features, for instance? Would the presence of defects have an influence in any of the other results presented?
Materials, line 87-88: What is meant here by structure? It’s unclear whether sentence refers to the full-size wooden artifact or the cellular structure that is shown in Figure 1. Please specify and/or add the relevant length scale.
Line 115: Please add a brief statement describing how this moisture analyzer measures humidity. For instance, is it a gravimetric method?
Line 182: Please mention the corrections applied to the spectra.
Line 255: Please clarify which cellulose varies between 56 to 62% or identify the method by which the percentage was obtained.
Author Response
Response to Reviewer 1 Comments
Very interesting paper. I've listed below my suggested revisions. I also suggest to carefully re-read the manuscript to correct typos and minor English grammar errors.
Thank you for your precise and detailed comments that helped to improve the manuscript! Authors highly appreciate your work.
Point 1: Abstract, I suggest the word assays is removed.
Response 1: Authors agree with this comment. The word is removed.
Point 2: Abstract, I believe there is a missing word in this sentence. Please verify, perhaps the following correction is needed: “also confirmed by FTIR”
Response 2: Authors agree and corrected this part.
Point 3: Introduction, Pleas correct the typo, it should be 14th century.
Response 3: Authors agree with that and corrected.
Point 4: Introduction, suggest replacing the word included for involved.
Response 4: Authors agree and replaced.
Point 5: Introduction, It is not clear what is meant by the following sentence: “Growing of each one has its own role in this complex system.”
Response 5: Thank you for your point. In this case, we replaced the existing sentence with the following one: “…of different species are involved, such as insects, calms, nematodes, fungi, and bacteria. The growth and development of each of the afore-mentioned species has its own role in this complex system.”
Point 6: Introduction, Please paraphrase the last sentence to clarify what is meant by conservation status. Do you mean that a non-destructive assessment was not needed due to the specimens being already decayed? Also, I suggest, the word stored is removed from this sentence since the samples were buried at a depth of 10m but not necessarily stored in soil.
Response 6: Thank you for your comment. The word stored is removed and the sentence is changed with this sentence: “Such samples have resisted severe conditions in the wet ground for around 2000 years; hence, their conservation status is little affected.”
Point 7: Materials, Please elaborate why it was necessary to compare the cellular structure of freshly cut elm wood that had no defects. How would the presence of reaction wood, or a steep annual growth ring change the anatomical features, for instance? Would the presence of defects have an influence in any of the other results presented?
Response 7: Authors agree and added this part: “Since knots and other abnormal parts of wood contain the highest amount of lignin and the lowest amount of cellulose comparing to other zones, for our tests, defect-free specimens (no knots, cracks, or wood reaction, and annual growth rings slope < 5°) were selected.”
Point 8: Materials, What is meant here by structure? It’s unclear whether sentence refers to the full-size wooden artifact or the cellular structure that is shown in Figure 1. Please specify and/or add the relevant length scale.
Response 8: Thank you for the comment. The sentence change to “The anatomical structure of this specimen was completely altered for a long time due to the aging of the wood as well as its burial in the soil.” Also, the images are corrected.
Point 9: Please add a brief statement describing how this moisture analyzer measures humidity. For instance, is it a gravimetric method?
Response 9: Thank you for the comment. This information added to this part: “In our case, the moisture analyzer measures moisture thermogravimetrically. Approximately 1 g of each sample were placed into the device. Thermogravimetric moisture analysis defines moisture as the loss of mass observed when the sample is heated and is based, in theory, on the vaporization of water during the drying process.”
Point 10: Please mention the corrections applied to the spectra.
Response 10: Authors agree and mentioned the baseline-corrected into the text.
Point 11: Please clarify which cellulose varies between 56 to 62% or identify the method by which the percentage was obtained.
Response 11: Thank you for the comment. In the text is mentioned the cellulose content of Ulmus species varies between 56-62%.

Reviewer 2 Report
The submitted paper substantially lacks in presentation and interpretation of the results. The flow of the results is not optimal. For the improvement, please check the following comments:
Materials and methods:
Line 79: the subsection name should be rewritten, maybe as: raw materials and preparation of samples or something like that. The existing name seems like incomplete and unfinished
Line 91: Are the Figures 1a and 1b a result of your own (this) research, if are, I think they should be placed in the result section.
Lines 87-88: “The structure of this specimen was completely altered for a long time due to the aging of the wood as well as its burial in the soil” – was the structure altered only due to aging? What about other environmental factors and contributors, i.e. biological degradation, as your paper is related to? This statement is mentioned in the manuscript several times and it should be corrected.
Line 128: “After the extraction (process described in chapter 2.2.2.),…“ isn't it section 2.2.3?
Line 137: „…was put to a 250 mL glass beaker…“ – I think its more grammar appropriate „was placed into a 250 mL beaker“
Lines 155 – 159: Is this method for lignin content determination a standard method? If is, name which.
Line 193: 3.1. Biological degradation. I think the name of the section is misleading. You have presented the results of light microscopy. You can talk about biological degradation in the conclusion and by considering all the given results. In this section you don’t have any results dealing with number of isolated bacteria or the micrographs of bacterial on the samples. The name of the section should be rewritten.
Lines 196-197: “specimens were buried 10 meters deep in the soil and the amount of oxygen there is very low“ – if you don't have any result of the oxygen measurement from the 10m depth of soil, than you cannot claim with certainty that the concentrations of the oxygen are low. Perhaps, there is no oxygen at all. It can any be an assumption.
Line 277: in figure 3, the most important peaks should be inserted.
Line 283: there are no OH classes, there are OH groups or OH vibrational bands
Line 284: not OH peaks, when analysing the FTIR spectra, we talk about OH vibrational bands.
Line 284-285: “Changes in OH peaks can be caused by the absorption of ambient moisture from wood specimens during measurement. “ It is questionable statement. The samples should be analysed in the controlled atmosphere, i.e. the samples should be analysed in the same conditions. Thus, the humidity from the environment can affect the measurements equally.
Line 287 – 288: „The peak at 1734 cm-1 belonging to the C=O group in hemicelluloses in archaeological
specimen does exhibit a significant decrease to the reference spectrum.“ I can’t see any changes for this part of the spectrum. There is no change in the peak shape, only in the intensity. But result in the intensity shift can be related to the factor of measurement performance. Additionally, only significant changes in the shape of the bands can serve as a good tool for assessment of any process that causes changes in the structure.
Lines 288-289: “Cellulose and hemicellulose are the first elements that are degraded during aging and with their degradation, the amount of lignin increases “. This sentence is questionable. How can lignin amount can increase during degradation? It can only remain constant or degraded. There is no process that can cause a cellulose/hemicellulose transfer to lignin.
Lines 291-292: “the increase in bands belonging to the lignin, might be caused by the leaching of hemicellulose and cellulose in the naturally aged wood [19].“ Which are the mentioned bands, in which wave numbers?
Discussion and conclusion:
The authors paid attention to the biological degradation of elm tree, and when comparing and explaining the results, they have used terminology and conclusions related to wood aging. Biodegradation and aging process are not the same so please do not mix it. Furthermore, a sample of wood was buried in the ground at a depth of 10 meters. What about soil moisture, did you measure it? Moisture can also play a significant role in degradation process, especially in low oxygen conditions. Also, the authors confidently talk about lower oxygen concentrations in the soil without conducting measurements. It is possible that the soil is completely anaerobic. Furthermore, claims that lignin concentrations increase with time are questionable. Lignin either degrades slowly or does not degrade, especially in condition where oxygen concentrations are low. Accordingly, the lignin / cellulose ratio in the sample can be increased, only the ratio but not the amount. I believe that these claims should be corrected and further literature on anaerobic (with anaerobic bacteria or facultative anaerobes) and aerobic degradation (bacteria or fungus) of wood in soil conditions should be researched. In view of all the above, the abstract and conclusion should be reformulated.
Author Response
Response to Reviewer 2 Comments
The submitted paper substantially lacks in presentation and interpretation of the results. The flow of the results is not optimal. For the improvement, please check the following comments:
Thank you for your precise and detailed comments that helped to improve the manuscript! Authors highly appreciate your assistance.
Point 1: the subsection name should be rewritten, maybe as: raw materials and preparation of samples or something like that. The existing name seems like incomplete and unfinished.
Response 1: Authors agree with this comment. The subsection changed to raw materials and preparation of samples.
Point 2: Are the Figures 1a and 1b a result of your own (this) research, if are, I think they should be placed in the result section.a
Response 2: Authors agree and corrected this part.
Point 3: “The structure of this specimen was completely altered for a long time due to the aging of the wood as well as its burial in the soil” – was the structure altered only due to aging? What about other environmental factors and contributors, i.e. biological degradation, as your paper is related to? This statement is mentioned in the manuscript several times and it should be corrected.
Response 3: Authors agree with that and corrected. The anatomical structure of this specimen was completely altered for a long time due to the environmental factors such as humidity and temperature also biologically degrading agents such as fungi and bacteria of the wood as well as its burial in the soil.
Point 4: “After the extraction (process described in chapter 2.2.2.),…“ isn't it section 2.2.3?
Response 4: Thank you for your comment. This part corrected to 2.2.3. in the text.
Point 5: „…was put to a 250 mL glass beaker…“ – I think its more grammar appropriate „was placed into a 250 mL beaker“
Response 5: Authors agree and corrected the text.
Point 6: Is this method for lignin content determination a standard method? If is, name which.
Response 6: Thank you for your comment. The 1% NaOH solubility test is good for the determination of lignin, and is well suitable for wood and pulp, and dissolves some readily soluble polyoses, yet it is also documented that it also dissolves degraded cellulose (Hon and Shiraishi 2001), which can be an issue in the case of historial artefacts, thus rendering the cellulose content of recent (intact) and of the „old” wood not comparable. So, in our opinion it would not add any significant information to the present results. In the current research, we measured holocellulose and alpha cellulose content and calculated hemicelluloses as the difference using a method Described by Rowell et al. (2012)
Determination of the structural polymer content as well as the extractive content of wood, subfossile wood and fossile wood are done in many different ways. Gnerally, none of the methods is perfect the sum of measuring all of the components will not equal 100 %. See for example Wentzel et al. 2018, Table2/Sluiter et al 2012. The use of subtracting known compositions from 100 to estimate e.g hemicellulose content (Fodor et al. 2018) or cellulose content () in native- modified- and fossile wood has already been applied and documented. We focused on showing differences between chemical composition of native and historical elm samples and in this regard using and applying the same methods protocol consequently will reflect the difference between the chemical composition of the samples in our opinion correctly.
References:
Fodor et al.: Effect of acetylation on the chemical composition of hornbeam (Carpinus betulus L.) in relation with the physical and mechanical properties. Wood Materials Science and Engineering, 2017. http://dx.doi.org/10.1080/17480272.2017.1316773
Hon, D.N.S and Shiraishi N: Wood and Cellulosic Chemistry, Marcel Dekker, 2001. p. 304.
Marynowski et al.: Occurrence and significance of mono-, di- and anhydrosaccharide
biomolecules in Mesozoic and Cenozoic lignites and fossil wood. 2018. https://doi.org/10.1016/j.orggeochem.2017.11.008
Rowell, R.M.: Handbook of Wood Chemistry and Wood Composites, CRC Press, 2012. p33-66.
Sluiter et al.: Determination of Structural Carbohydrates and Lignin in Biomass Laboratory Analytical Procedure (LAP). 2012.
Wentzel et al.: Relation of chemical and mechanical properties of Eucalyptus nitens wood thermally modified in open and closed systems. 2018. https://doi.org/10.1080/17480272.2018.1450783
Point 7: 3.1. Biological degradation. I think the name of the section is misleading. You have presented the results of light microscopy. You can talk about biological degradation in the conclusion and by considering all the given results. In this section you don’t have any results dealing with number of isolated bacteria or the micrographs of bacterial on the samples. The name of the section should be rewritten.
Response 7: Authors agree, and name of the section has changed to Evaluation of bacterial attack by Light Microscopy.
Point 8: “specimens were buried 10 meters deep in the soil and the amount of oxygen there is very low“ – if you don't have any result of the oxygen measurement from the 10m depth of soil, than you cannot claim with certainty that the concentrations of the oxygen are low. Perhaps, there is no oxygen at all. It can any be an assumption.
Response 8: Authors agree with this comment. Due to the age of the specimen and the possibility of change during this period, we do not have information about the amount of oxygen in the area. As you said, there was probably no oxygen at all. For this reason, the sentences related to this discussion have been changed. Instead of claiming with certainty that concentrations are low we assert a possibility of low amount of oxygen present in that conditions.
Point 9: in figure 3, the most important peaks should be inserted.
Response 9: Thank you for the comment. The most important peaks were inserted in figure 3.
Point 10: there are no OH classes, there are OH groups or OH vibrational bands
Response 10: Authors agree and corrected in the text.
Point 11: not OH peaks, when analysing the FTIR spectra, we talk about OH vibrational bands.
Response 11: Thank you for the comment. Corrected in the text of manuscript.
Point 12: “Changes in OH peaks can be caused by the absorption of ambient moisture from wood specimens during measurement. “It is questionable statement. The samples should be analysed in the controlled atmosphere, i.e. the samples should be analysed in the same conditions. Thus, the humidity from the environment can affect the measurements equally.
Response 12: Thank you for your point. All samples in this work were analysed at the same conditions of temperature and humidity.
Point 13: „The peak at 1734 cm-1 belonging to the C=O group in hemicelluloses in archaeological specimen does exhibit a significant decrease to the reference spectrum.“ I can’t see any changes for this part of the spectrum. There is no change in the peak shape, only in the intensity. But result in the intensity shift can be related to the factor of measurement performance. Additionally, only significant changes in the shape of the bands can serve as a good tool for assessment of any process that causes changes in the structure.
Response 13: Thank you for your comment. Yes, we agree with you. Only intensity in the peak of 1734 cm-1 had changed in historical samples. According to the response in the previous comment, all samples in this study were analysed under the same conditions. This sentence has changed in the text.
Point 14: “Cellulose and hemicellulose are the first elements that are degraded during aging and with their degradation, the amount of lignin increases “. This sentence is questionable. How can lignin amount can increase during degradation? It can only remain constant or degraded. There is no process that can cause a cellulose/hemicellulose transfer to lignin.
Response 14: Thank you for your comment. Writing the amount in this part was wrong and the percentage should have been written. The amount of lignin does not increase, but as the amount of holocellulose decreases, the percentage of lignin increases.
Point 15: “the increase in bands belonging to the lignin, might be caused by the leaching of hemicellulose and cellulose in the naturally aged wood [19].“ Which are the mentioned bands, in which wave numbers?
Response 15: Thank you for your comment. The peak numbers are added and the sentence changed to: “The increase in bands between 1645 and 1237 cm-1 belonging to the lignin, might be caused by the leaching of hemicellulose and cellulose in the naturally aged wood [19].”
Point 16: Discussion and conclusion:
The authors paid attention to the biological degradation of elm tree, and when comparing and explaining the results, they have used terminology and conclusions related to wood aging. Biodegradation and aging process are not the same so please do not mix it. Furthermore, a sample of wood was buried in the ground at a depth of 10 meters. What about soil moisture, did you measure it? Moisture can also play a significant role in degradation process, especially in low oxygen conditions. Also, the authors confidently talk about lower oxygen concentrations in the soil without conducting measurements. It is possible that the soil is completely anaerobic. Furthermore, claims that lignin concentrations increase with time are questionable. Lignin either degrades slowly or does not degrade, especially in condition where oxygen concentrations are low. Accordingly, the lignin / cellulose ratio in the sample can be increased, only the ratio but not the amount. I believe that these claims should be corrected and further literature on anaerobic (with anaerobic bacteria or facultative anaerobes) and aerobic degradation (bacteria or fungus) of wood in soil conditions should be researched. In view of all the above, the abstract and conclusion should be reformulated.
Response 16: According to the changes made in the manuscript, the abstract, as well as the conclusion, changed.

Round 2
Reviewer 2 Report
I have no comments.
This manuscript is a resubmission of an earlier submission. The following is a list of the peer review reports and author responses from that submission.